# Vietnamese *Dalbergia tonkinensis*: A Promising Source of Mono- and Bifunctional Vasodilators

**DOI:** 10.3390/molecules27144505

**Published:** 2022-07-14

**Authors:** Nguyen Manh Cuong, Ninh The Son, Ngu Truong Nhan, Yoshiyasu Fukuyama, Amer Ahmed, Simona Saponara, Alfonso Trezza, Beatrice Gianibbi, Ginevra Vigni, Ottavia Spiga, Fabio Fusi

**Affiliations:** 1Institute of Natural Products Chemistry, Vietnam Academy of Science and Technology (VAST), 18 Hoang Quoc Viet, Caugiay, Hanoi 122100, Vietnam; 2Institute of Chemistry, Vietnam Academy of Science and Technology (VAST), 18 Hoang Quoc Viet, Caugiay, Hanoi 122100, Vietnam; yamantson@gmail.com; 3Faculty of Science and Technology, Tay Nguyen University, 567 Le Duan, Ea Tam, Buon Ma Thuot 630000, Vietnam; ntnhan@ttn.edu.vn; 4Faculty of Pharmaceutical Sciences, Tokushima Bunri University, Tokushima 770-8514, Japan; fukuyama@ph.bunri-u.ac.jp; 5Dipartimento di Scienze della Vita, Università di Siena, Via A. Moro 2, 53100 Siena, Italy; aa.biotechiub@gmail.com (A.A.); simona.saponara@unisi.it (S.S.); 6Dipartimento di Biotecnologie, Chimica e Farmacia, Università di Siena, Via A. Moro 2, 53100 Siena, Italy; alfonso.trezza2@unisi.it (A.T.); beatrice.gianibbi@student.unisi.it (B.G.); ginevra.vigni@student.unisi.it (G.V.); ottavia.spiga@unisi.it (O.S.)

**Keywords:** biochanin A, Ca_V_1.2 channel, *Dalbergia tonkinensis* Prain, K_Ca_1.1 channel, vasodilation

## Abstract

Hypertension is a risk factor for cardiovascular diseases, which are the main cause of morbidity and mortality in the world. In the search for new molecules capable of targeting K_Ca_1.1 and Ca_V_1.2 channels, the expression of which is altered in hypertension, the in vitro vascular effects of a series of flavonoids extracted from the heartwoods, roots, and leaves of *Dalbergia tonkinensis* Prain, widely used in traditional medicine, were assessed. Rat aorta rings, tail artery myocytes, and docking and molecular dynamics simulations were used to analyse their effect on these channels. Formononetin, orobol, pinocembrin, and biochanin A showed a marked myorelaxant activity, particularly in rings stimulated by moderate rather than high KCl concentrations. Ba^2+^ currents through Ca_V_1.2 channels (I_Ba1.2_) were blocked in a concentration-dependent manner by sativanone, 3′-*O*-methylviolanone, pinocembrin, and biochanin A, while it was stimulated by ambocin. Sativanone, dalsissooside, and eriodictyol inhibited, while tectorigenin 7-*O*-[β-D-apiofuranosyl-(1→6)-β-D-glucopyranoside], ambocin, butin, and biochanin A increased I_KCa1.1_. In silico analyses showed that biochanin A, sativanone, and pinocembrin bound with high affinity in target-sensing regions of both channels, providing insight into their potential mechanism of action. In conclusion, *Dalbergia tonkinensis* is a valuable source of mono- and bifunctional, vasoactive scaffolds for the development of novel antihypertensive drugs.

## 1. Introduction

Cardiovascular diseases, a group of disorders that affect blood vessels and the heart, represent the leading cause of death worldwide, which resulted in 17.8 million deaths in 2017 [1]. They are generally preceded by many simultaneous modifiable risk factors (e.g., hypertension, hyperglycaemia, obesity, and oxidative stress) that may be associated with genetic factors and/or unhealthy lifestyles (e.g., physical inactivity, smoking, inappropriate diet, mental stress, and ageing; [2]. Approximately 1.4 billion people worldwide suffer from hypertension [3], a well-known mosaic of pathophysiological disturbances, the successful treatment of which is possible given the availability of multiple antihypertensive drug classes (characterised by various mechanisms of action) with limited side effects. This guarantees a complementary pharmacological treatment. As an alternative, bi- or multifunctional drugs, capable of simultaneously targeting more than one mechanism underpinning hypertension, may represent an ideal therapeutic regimen, perhaps accompanied by fewer unwanted effects.

Although numerous antihypertensive drugs exist, neither therapy based on their association nor fixed-dose combinations have provided satisfactory results due to the fact of poor compliance and variable and unpredictable pharmacokinetics and pharmacodynamics in different patients. In this context, multitarget agents, which combine two or more pharmacophores capable of modulating different mechanisms involved in the pathogenesis of hypertension, play a fundamental role. They are single-chemical entities, which reduces the risk of drug–drug interactions and allow for a simpler prediction of patient pharmacokinetics and pharmacodynamics.

In recent years, the pharmaceutical industry and academic research have, again, directed their attention to the natural world, which has always represented an important source of active agents. This occurred following the failure of libraries of synthetic compounds and computational chemistry, which were thought to lead to the discovery of numerous new drugs. Currently, molecules of natural origin play an important role in the development of novel drugs: they are compounds that have a unique chemical diversity that translates into interesting pharmacological properties. Their effectiveness is due to the fact of their three-dimensional chemical structure and a complex steric property (the so-called “privileged structure”), which also offers an advantage in terms of selectivity for molecular targets. This makes natural molecules multitarget compounds and, therefore, a valuable starting point for the development of new drugs, especially in the field of antihypertensive therapy, where approximately 64% of drugs are derived from natural products [4].

In this scenario, the most interesting insights are provided by traditional medicine that makes use of natural extracts to treat patients otherwise not reachable by “conventional” medicine. The Asian continent plays a fundamental role, with China certainly in the foreground but also smaller countries, such as Vietnam, where the use of over 4000 plants in traditional medicine is essential for health. These include *Dalbergia* species, belonging to a large genus of trees, shrubs, lianas, and woody climbers of the Fabaceae family (Leguminosae), the extracts of which are widely used for the treatment of cardiovascular disease [5,6]. *Dalbergia odorifera*, a Chinese herbal medicine, has been used for promoting blood circulation, relieving pain, and eliminating blood stasis [7]. Thus far, more than 50 compounds have been isolated and identified from *Dalbergia tonkinensis* Prain, local name “Sưa Đỏ”, one of 27 *Dalbergia* species distributed in Vietnam; these belong to the classes of flavonoids, polyphenols, sesquiterpenes, arylbenzofurans, and quinones [8,9,10,11,12,13]. Flavonoids, in particular, are the main chemical components extracted from duramen [8,10,11,12,14]. Accumulating evidence obtained from in vitro and in vivo research, clinical trials, and epidemiological studies point to flavonoids as beneficial tools capable of improving cardiovascular health and ameliorating the risk factors associated with cardiovascular diseases [15]. As a previous study demonstrated the vasorelaxant activity of an open-ring neoflavonoid, namely, *R*-(-)-3′-hydroxy-2,4,5-trimethoxydalbergiquinol [16], extracted from the heartwood of *D. tonkinensis*, this work aimed to evaluate a series of 12 flavonoids (including flavanones, isoflavanones, and isoflavones; Figure 1) extracted from the heartwoods, roots, and leaves of *D. tonkinensis* as potential vasodilators endowed with K_Ca_1.1 channel-stimulating and/or Ca_V_1.2 channel-blocking activity. The results indicate that *D. tonkinensis* can be considered a valuable source of vasoactive compounds, with biochanin A, in particular, being the most interesting structure, which could be used as a scaffold for the development of novel antihypertensive drugs.

## 2. Materials and Methods

### 2.1. Plant Material

Heartwoods of *Dalbergia tonkinensis* Prain were collected in Quang Binh Province (2016), and whole plants (i.e., leaves, heartwoods, and roots) over ten years old were collected in Daklak Province (2016), Vietnam. The plant was identified by Botanist, Nguyen Quoc Binh, Vietnam National Museum of Nature, VAST, Hanoi, Vietnam. Voucher specimens (C-575, Quang Binh; C-561 and C-612, Daklak, Vietnam) were deposited at the Department of Bioactive Products, Institute of Natural Products Chemistry, VAST, Hanoi, Vietnam.

### 2.2. General Experimental Procedures

^1^H-NMR (500 MHz) and ^13^C-NMR (125 MHz) spectra were measured on a Bruker Avance 500 MHz spectrometer. ESI-MS was obtained from a Varian FT-MS spectrometer and MicroQ-TOF III (Bruker Daltonics, Ettlingen, Germany). Column chromatography was carried out on silica gel (Si 60 F254, 40–63 mesh, Merck, St. Louis, MO, USA). All solvents were redistilled before use. Precoated thin-layer chromatography (TLC) plates (Si 60 F254) were used for analytical purposes. Compounds were visualised under UV light (254 and 365 nm) and by spraying plates with 10% H_2_SO_4_ followed by heating with a heat gun.

### 2.3. Extraction and Isolation

*D. tonkinensis* dried leaves (2 kg) were extracted using a Soxhlet extractor (methanol: 10 L, 4.5 h) to yield a black crude extract (300 g), which was suspended in methanol–water 1:1 (*v*/*v*) and partitioned with chloroform and ethyl acetate to obtain the appropriate fractions.

The chloroform fraction (LC, 60.5 g) was fractionated using column chromatography over silica gel (chloroform–methanol, 99:1 (*v*/*v*)) to give 8 fractions (LC1-LC8). Fraction LC1 (15.2 g) was purified over silica gel column chromatography (*n*-hexane–acetone, 10:1 (*v*/*v*)) to successfully yield biochanin A (45.7 mg). Orobol (15.6 mg) was obtained from fraction LC6.

The water-residual fraction (LW, 95.5 g) was subjected to Diaion HP-20 column chromatography, eluted with water–methanol (1:0, 1:3, and 0:1, *v*/*v*), providing 3 fractions (LW1-LW3). LW2 (4.57 g) was chromatographed on silica gel column chromatography (chloroform–methanol–water, 3:1:0.1 (*v*/*v*/*v*)) to yield dalsissooside (20.6 mg) and 9 fractions (LW21-LW29). Fraction LW28 (1.9 g) was then separated by HPLC (Cosmosil 5C_18_-AR-II column, methanol–water ratio of 7:3 (*v*/*v*, 1.5 mL/min), UV 254 nm) to yield ambocin (4.6 mg).

*D. tonkinensis* dried roots (2 kg) were extracted using a Soxhlet extractor (ethanol: 10 L, 4.5 h) to yield a black crude extract (200 g), which was resuspended in a methanol–water ration of 1:1 (*v*/*v*) and partitioned with dichloromethane and ethyl acetate to produce the corresponding fractions. The ethyl acetate fraction (RE, 30.5 g) was separated by column chromatography using a chloroform–methanol–water ration of 3:1:0.1 (*v*/*v*/*v*) providing 7 fractions (RE1-RE7). Tectoridin (5.1 mg) and tectorigenin-7-*O*-[β-D-apiofuranosyl-(1→6)-β-D-glucopyranoside (25.2 mg) were obtained from fraction RE4 (7.9 g) using a Sephadex LH-20 column (methanol–water, 1:1 (*v*/*v*)) [12].

Other flavonoids, namely, pinocembrin [8], 3’-*O*-methylviolanone [10], biochanin A, butin, eriodictyol [11], sativanone, and formononetin [14] were isolated from the heartwoods of *D. tonkinensis* as previously described.

### 2.4. Animals

All the study procedures were in strict accordance with the European Union Guidelines for the Care and the Use of Laboratory Animals (European Union Directive 2010/63/EU) and approved by the Animal Care and Ethics Committee of the University of Siena and the Italian Department of Health (7DF19.N.TBT). Male Wistar rats (300–350 g) were purchased from Charles River Italia (Calco, Italy) and maintained in an animal house facility at 25 ± 1 °C with a 12:12 h dark–light cycle and access to standard chow diet and water ad libitum. Animals were anaesthetised with an isoflurane (4%) and O_2_ gas mixture using Fluovac equipment (Harvard Apparatus, Holliston, MA, USA), decapitated, and exsanguinated. Thoracic aorta and tail main artery were immediately isolated and placed in physiological solutions (i.e., the modified Krebs–Henseleit solution (KHS) and an external solution, respectively) and prepared as detailed below.

### 2.5. Functional Experiments

#### 2.5.1. Aorta Rings Preparation

The thoracic aorta was gently cleansed of adipose and connective tissues and cut into 3 mm wide rings. They were mounted in organ baths between two parallel, L-shaped, stainless-steel hooks, one fixed in place and the other connected to an isometric transducer. Rings were allowed to equilibrate for 60 min in KHS (composition in mM: 118 NaCl, 4.75 KCl, 1.19 KH_2_PO_4_, 1.19 MgSO_4_, 25 NaHCO_3_, 11.5 glucose, and 2.5 CaCl_2_ and gassed with a 95% O_2_–5% CO_2_ gas mixture to create a pH of 7.4) under a passive tension of 1 g. During this equilibration period, the solution was changed every 15 min. Isometric tension was recorded using a digital PowerLab data acquisition system (PowerLab 8/30; ADInstruments). Ring viability was assessed by recording the response to 0.3 µM phenylephrine and 60 mM KCl [17]. Where needed, the endothelium was removed by gently rubbing the lumen of the ring with a forceps tip. This procedure was validated by adding 10 µM acetylcholine at the plateau of phenylephrine-induced contraction: a relaxation greater than 70% or less than 10% denoted the presence or absence of functional endothelium, respectively [18].

#### 2.5.2. Effect of Dalbergia Isolates on KCl-Induced Contraction

Endothelium-denuded aorta rings were precontracted electromechanically with 25 mM or 60 mM KCl. Once the contraction reached a plateau, drugs were added cumulatively into the organ bath to assess their vasodilating activity. At the end of the concentration–response curve, 1 µM nifedipine followed by 100 µM sodium nitroprusside was added to test the functional integrity of smooth muscle. Vasodilation was calculated as a percentage of the contraction induced by KCl (taken as 100%).

### 2.6. Cell Isolation Procedure

The tail main artery was dissected free of its connective tissue, and smooth muscle cells were freshly isolated under the following conditions. A 5 mm long piece of artery was incubated at 37 °C for 40–45 min in 2 mL of 0.1 mM Ca^2+^ external solution (consisting of (in mM): 130 NaCl, 5.6 KCl, 10 HEPES, 20 glucose, 1.2 MgCl_2_, and 5 Na-pyruvate; pH 7.4) containing 20 mM taurine, which replaced an equimolar amount of NaCl, 1.35 mg/mL collagenase (type XI), 1 mg/mL soybean trypsin inhibitor, and 1 mg/mL BSA. This solution was gently bubbled and stirred with a 95% O_2_–5% CO_2_ gas mixture as previously described [19]. Cells stored in 0.05 mM Ca^2+^ external solution containing 20 mM taurine and 0.5 mg/mL BSA at 4 °C under normal air were used for experiments within two days after isolation [20].

### 2.7. Whole-Cell Patch-Clamp Recordings

An Axopatch 200B patch-clamp amplifier (Molecular Devices Corporation, Sunnyvale, CA, USA) was used to generate and apply voltage pulses to the clamped cells and record the corresponding membrane currents. Recording electrodes were pulled from borosilicate glass capillaries (WPI, Berlin, Germany) and fire-polished to obtain a pipette resistance of 2–5 MΩ when filled with an internal solution. At the beginning of each experiment, the junction potential between the pipette and bath solution was electronically adjusted to zero. Current signals, after compensation for whole-cell capacitance and series resistance (between 70% and 75%), were low-pass filtered at 1 kHz and digitised at 3 kHz before being stored on a computer hard disk. Electrophysiological responses were tested at room temperature (20–22 °C) [21].

### 2.8. Ba^2+^ Current through Ca_V_1.2 Channel (I_Ba1.2_) Recordings

Cells were continuously superfused with an external solution containing 0.1 mM Ca^2+^ and 30 mM tetraethylammonium (TEA) using a peristaltic pump (LKB 2132, Bromma, Sweden) at a flow rate of 400 µL/min. The conventional whole-cell patch-clamp method was employed to voltage-clamp smooth muscle cells. The internal solution (pCa 8.4) consisted of (in mM): 100 CsCl, 10 HEPES, 11 EGTA, 2 MgCl_2_, 1 CaCl_2_, 5 Na-pyruvate, 5 succinic acid, 5 oxaloacetic acid, 3 Na_2_ATP, and 5 phosphocreatine; the pH was adjusted to 7.4 with CsOH. I_Ba1.2_, recorded in an external solution containing 30 mM TEA and 5 mM Ba^2+^, was elicited with 250 ms clamp pulses (0.067 Hz) to 0 mV from a V_h_ of −50 mV. Ba^2+^ was used in place of Ca^2+^ as the charge carrier to increase the current density that, in rat tail artery myocytes, is usually around 1 pA/pF (corresponding to an average current amplitude of 30–50 pA; see [22]. Data were collected once the current amplitude stabilised (usually 7–10 min after the whole-cell configuration was obtained). Under these conditions, the current, which did not run down during the following 40 min [23], was carried almost entirely by Ca_V_1.2 channels [22]. The K^+^ currents were blocked with 30 mM TEA in the external solution and Cs^+^ in the internal solution. Current values were corrected for leakage and residual outward currents using 10 µM nifedipine, which completely blocked I_Ba1.2_. The osmolarity of the 30 mM TEA- and 5 mM Ba^2+^-containing external solution (320 mosmol, adjusted with NaCl if required) and that of the internal solution (290 mosmol) were measured with an osmometer (Osmostat OM 6020, Menarini Diagnostics, Florence, Italy).

### 2.9. K^+^ Current through K_Ca_1.1 Channel (I_KCa1.1_) Recordings

I_KCa1.1_ (registration period 500 ms) was measured over a range of test potentials from −20 to 70 mV from a V_h_ of −40 mV. This V_h_ limited the contribution of voltage-dependent K^+^ channels to the overall whole-cell current. Data were collected once the current amplitude stabilised (usually 6–10 min after the whole-cell configuration was obtained). I_KCa1.1_ did not run down during the following 20–30 min under the present experimental conditions [24]. The external solution for I_KCa1.1_ recordings contained (in mM): 145 NaCl, 6 KCl, 10 glucose, 10 HEPES, 5 Na-pyruvate, 1.2 MgCl_2_, 0.1 CaCl_2_, and 0.003 nicardipine (pH 7.4). The internal solution contained (in mM): 90 KCl, 10 NaCl, 10 HEPES, 10 EGTA, 1 MgCl_2_, and 6.41 CaCl_2_ (pCa 7.0; pH 7.4). The osmolarity of the external and internal solutions was 310 mosmol and 265 mosmol, respectively. The current–voltage relationships were calculated based on the values recorded over the last 400 ms of each test pulse (leakage corrected). The I_KCa1.1_ was isolated from the other currents as well as corrected for leakage using 1 mM TEA, a specific blocker of K_Ca_1.1 channels [25].

### 2.10. Drugs and Chemicals

The chemicals used included collagenase (type XI), trypsin inhibitor, BSA, TEA chloride, HEPES, taurine, phenylephrine, acetylcholine, nifedipine, nicardipine (Sigma Chimica, Milan, Italy), and sodium nitroprusside (Riedel-De Haën AG, Seelze-Hannover, Germany). All other substances were of analytical grade and used without further purification. Phenylephrine was solubilised in 0.1 M HCl. Nifedipine and nicardipine, dissolved directly in ethanol, and *Dalbergia* isolates, dissolved directly in DMSO, were diluted at least 1000 times before use. Control experiments confirmed that no response was observed in vascular preparations when DMSO or ethanol, at the final concentration used in the above dilutions (0.1%, *v*/*v*), was added alone (data not shown).

### 2.11. Statistical Analysis

Analysis of the data was accomplished using pClamp 9.2.1.8 software (Molecular Devices Corporation, Sunnyvale, CA, USA), LabChart 7.3.7 Pro (PowerLab; ADInstruments, Castle Hill, Australia), and GraphPad Prism version 5.04 (GraphPad Software Inc., San Diego, CA, USA). Data are reported as the mean ± SEM; *n* is the number of cells or rings analysed (indicated in parentheses), isolated from at least three animals. Statistical analyses and significance, as measured by repeated measures ANOVA (followed by Dunnett’s post hoc test) or the Student’s *t*-test for paired samples (two-tailed), were obtained using GraphPad Prism version 5.04 (GraphPad Software Inc.). In all comparisons, *p* < 0.05 was considered significant. The pharmacological response to drugs, described in terms of potency (IC_50_ value, i.e., the drug concentration decreasing the maximum response by 50%) and efficacy (E_max_ value, i.e., the maximum response achieved with the highest concentration tested), was obtained by nonlinear regression analysis. For I_Ba1.2_ recordings, the bottom of the concentration–response curve to morin − 1 was constrained to the value recorded in the presence of the specific blocker nifedipine, taken as a 0% current amplitude.

### 2.12. Structural Resource

The *Rattus norvegicus* Ca_V_1.2 channel’s subunit α_1C_’s 3D structure was achieved with a homology modelling procedure as described by Trezza et al. [26]. The rabbit inactive Ca_V_1.1 channel’s 3D structure (PDB code 6JPA) in complex with its blocker verapamil, recently obtained through cryoEM by Zhao et al. [27], was selected as a template to build the Ca_V_1.2 3D model and explore the structural and energetic features of the compounds binding to the channel. The primary structure of the *Rattus norvegicus* K_Ca_1.1 channel was downloaded from the UniProt Database [28] (UniProt ID—Q62976-), and it was used as a query sequence for a multiple sequences alignment (MSA) carried out by Clustal Omega, implemented in PyMOD3.0 [29], choosing the Protein Data Bank (pdb) as a database; all algorithm parameters were used by default. As evidenced by the MSA results, the Cryo-EM structure of the Ca^2+^-bound hsSlo1-beta4 channel (in the open state) complex (PDB code: 6V22) was the best template [30], showing a cover and identity of 90.2% and 99.4%, respectively. Then, 6V22 was identified as a template to rebuild the 3D structure of the *Rattus norvegicus* K_Ca_1.1 channel, using the Modeller tool implemented in PyMOD3.0 [29]. The validity of the 3D structure was assessed using Ramachandran plot and PROCHECK analyses as previously described [31]. The 3D structures of biochanin A, sativanone, and pinocembrin (compound CIDs: 5280373, 13886678, and 68071, respectively) were downloaded in the sdf format using the PubChem database [32]. The compounds were sketched and then prepared by the LigPrep tool, assigning charges with Epik at pH 7.00 ± 1.00 [33]. The channel’s 3D model structures were converted into the pdbqt format as described in a previous work [34].

### 2.13. Docking and Classical Molecular Dynamics Simulations

To identify the potential binding pose of compounds on the Ca_V_1.2 and K_Ca_1.1 channels, in silico molecular docking and flexible sampling were applied using the glide standard precision (SP) protocol [35]. Input charges of compounds were retained, and amide bond conformations were allowed to vary. Strain correction terms were applied to the glide scoring function, and Epik state penalties were computed for the final docking score. All other options were set to default. The Receptor Grid Generation tool from Schrödinger 2019-2 was used to generate a box able to enclose all Ca_V_1.2 and K_Ca_1.1 binding-pocket residues. In brief, a box of 22 Å for each dimension was generated for the Ca_V_1.2 channel, enclosing a known blocker binding region of the protein [27]. To investigate the potential mechanism of action, a classical molecular dynamics (cMD) simulation of 100 ns was performed on the channel bound to the compounds as previously reported [26]. A box of 18 Å for each dimension was generated for the K_Ca_1.1 channel, enclosing a stimulator binding region as previously described [36]. To further investigate the channel/molecule interactions within the binding site, compounds were prepared as suggested by Semenya et al. [37], creating a phase database, and minimizing the output of 100 conformers per ligand. The structures were charged according to the Epik tool at pH 7.00 ± 1.00. Specified chirality and the 8 lowest energy stereoisomers (if present) were retained. Up to 4 low-energy 5- and 6-membered ring conformations were generated. All high-energy conformers/tautomers were discarded. Interaction network analyses and the energy contribution of the binding residues were evaluated through computational alanine-scanning mutagenesis, performed using the P.L.I.P. tool [38] and ABS scan [39], respectively. The GROMACS 2019.3 package was used to carry out and analyse the cMD trajectories [40]. PyMOL v2.5 was used as the molecular graphics system to generate the figures (The PyMOL Molecular Graphics System, version 2.5, Schrödinger, LLC, New York, NY, USA).

## 3. Results

Former phytochemical investigations of the heartwoods, leaves, and roots of *D. tonkinensis* resulted in the isolation of more than 35 flavonoids. The chemical structures of the compounds here investigated were previously identified by NMR analysis and literature comparison including the flavones: butin, eriodictyol [41], and pinocembrin [42]; the isoflavanones: sativanone and 3′-*O*-methylviolanone [10]; the isoflavones: biochanin A, formononetin [43], and orobol [44]; the isoflavone glycosides: ambocin [45], dalsissooside [46], tectoridin, and tectorigenin-7-*O*-[β-D-apiofuranosyl-(1→6)-β-D-glucopyranoside] [11] (Figure 1).

The compounds under study were assessed on aorta rings devoid of endothelium and precontracted by either 25 or 60 mM KCl, an assay to discriminate between Ca^2+^ antagonist and K^+^ channel opener agent [47].

Figure 2 shows the representative traces of the effects of biochanin A on the contraction induced by either 60 (panel A) or 25 mM KCl (panel B). The addition of cumulative concentrations of the compound on the plateau of KCl-induced active tone caused a concentration-dependent relaxation that was more evident when the degree of membrane depolarisation was reduced (i.e., at 25 mM KCl). As summarised in Figure 2C,D and Table 1, both isoflavanones (i.e., sativanone and 3′-*O*-methylviolanone), the flavanone pinocembrin and, to a lesser extent, the isoflavones, formononetin and biochanin A, were the most effective vasorelaxant agents on both types of contractions, efficacy being indirectly correlated to the extracellular concentration of KCl. Orobol and, to a lesser extent, eriodictyol, relaxed only 25 mM KCl-induced contractions. All other molecules assessed were ineffective (Table 1).

In a first series of experiments, the effects of biochanin A on I_Ba1.2_, recorded in rat tail artery myocytes following a depolarisation step to 0 mV from a V_h_ of −50 mV, were investigated. After I_Ba1.2_ reached a stable value, biochanin A was added at cumulative concentrations that produced a concentration-dependent decrease in the current amplitude (Figure 3A). The data obtained with biochanin A and the other *Dalbergia* isolates are summarised in Figure 3B–D and Table 1. Each subclass of flavonoids displayed at least one active blocker: pinocembrin among flavanones (panel B); both isoflavanones (i.e., sativanone and 3′-*O*-methylviolanone) assessed (panel C); biochanin A among the isoflavones (panel D). Only ambocin caused an approximately 20% increase in I_Ba1.2_ at the 10 µM concentration that, however, was not observed either at lower or higher concentrations.

I_Ba1.2_ evoked at 0 mV from a V_h_ of −50 mV activated and then declined with a time course that could be fitted by a monoexponential function. Only the two isoflavanones and biochanin A caused a significant concentration-dependent acceleration of the τ of inactivation (Figure 4). However, the τ of activation was not modified by any of the *Dalbergia* isolates (see as an example Figure 4).

A biophysical analysis was carried out to elucidate the interaction between the most effective compound—biochanin A—and Ca_V_1.2 channels. The current–voltage relationships (Figure 5A) show that 30 µM of biochanin A significantly decreased the peak inward current in the range of membrane potential values from −30 to 40 mV without, however, changing the maximum of the curve observed at 10 mV.

Figure 5B illustrates the time course of the effects of 30 µM biochanin A on the current recorded at 0.067 Hz from a V_h_ of −50 mV to a test potential of 0 mV. After I_Ba1.2_ reached steady values, the addition of biochanin A to the bath solution produced a gradual decrease in the current amplitude that reached a plateau in approximately 8 min. Washout of the compound brought the current amplitude back to 66% of the control value. Biochanin A-induced inhibition of I_Ba1.2_ was not affected by the membrane potential. In fact, when V_h_ shifted to −80 mV, the residual I_Ba1.2_ (38.6 ± 7.8% of the control, *n* = 5) was similar to that recorded at a V_h_ of −50 mV (25.9 ± 5.4% of the control, *n* = 5; *p* = 0.22).

The voltage dependence of biochanin A inhibition was further investigated by analysing the steady-state inactivation and activation curves for I_Ba1.2_. The steady-state activation curves (Figure 5C), calculated from the current–voltage relationships shown in panel A, were fitted with the Boltzmann equation. Biochanin A neither shifted the 50% activation potential (−3.72 ± 1.50 mV, control; −4.14 ± 2.15 mV, 30 µM biochanin A, *n* = 5; *p* = 0.77, Student’s *t*-test for paired samples) nor affected the slope factor (6.43 ± 0.40 mV and 6.34 ± 0.56 mV, respectively; *p* = 0.79). Conversely, biochanin A significantly shifted the steady-state inactivation curve to more negative potentials (Figure 5C). The 50% inactivation potential changed from −37.10 ± 3.76 mV (*n* = 6, control) to −50.23 ± 2.61 mV (30 µM biochanin A; *p* = 0.0051). The slope factor, however, was not affected by biochanin A (−12.58 ± 2.28 mV, −9.07 ± 0.83 mV, respectively; *p* = 0.14). The shift of the inactivation curve caused by 30 µM biochanin A led to a marked reduction in the Ba^2+^ window current that at −10 mV, for example, showed a relative amplitude of 0.02 compared to the relative amplitude of 0.06 observed under control conditions.

*Dalbergia* isolates were assessed for their effects on I_KCa1.1_. Under the conditions used in the present experiments, the outward current mostly consisted of iberiotoxin-sensitive I_KCa1.1_ [48]. Figure 6A shows the traces of I_KCa1.1_ elicited with clamp pulses to 70 mV from a V_h_ of −40 mV, under control conditions and after the cumulative addition of 30 and 100 µM biochanin A. The flavonoid caused a significant concentration-dependent stimulation of the current, which was observed in the range of membrane potential 20–70 mV (Figure 6B). The effects of the other compounds assessed are shown in Table 1. Both isoflavanones were almost ineffective or slightly reduced the current amplitude. Similar behaviour was shown by eriodictyol and pinocembrin, while the other flavanone, butin, at the maximal concentration tested, stimulated I_KCa1.1_ by 73%. Among the remaining isoflavones, only ambocin and tectorigenin 7-*O*-[β-D-apiofuranosyl-(1→6)-β-D-glucopyranoside] were capable of stimulating the current amplitude.

The overall effects of the compounds are summarised in Table 2.

To predict the potential binding pose of compounds on the homology model of the *Rattus norvegicus* Ca_V_1.2 channel’s α_1C_ subunit, a docking simulation was performed. The best-docked conformation of biochanin A, pinocembrin, and sativanone showed Gibbs free energy values (ΔG) of −6.2, −7.5, and −5.7, respectively. The compounds shared the same binding region, located close to the central pore (Figure 7).

The interaction network analysed by the P.L.I.P. tool demonstrated that biochanin A formed hydrophobic interactions with Leu-774 (S6-II), Leu-775 (S6-II), and Ala-1183 (S6-III); a hydrogen bond with Asn-771 (S6-II); two π stackings with Phe-778 (cytoplasmic) and Phe-1143 (pore forming) (Figure 7A). Sativanone triggered hydrophobic interactions with Phe-778 (cytoplasmic) and Phe-1143 (pore forming), and a hydrogen bond with Asn-771 (S6-II) (Figure 7B). Pinocembrin exhibited two hydrophobic interactions with Phe-730, Leu-777, Phe-778, Ile-1180, Ala-1183, and Phe-1184 (Figure 7C). To confirm the stability of both the protein structure and compound binding pose and to define a potential mechanism of action, a classical molecular dynamics (cMD) simulation of 100 ns was performed for the Ca_V_1.2 channel in a complex with biochanin A, sativanone, and pinocembrin. Root mean square deviation (RMSD) performed on the backbone of each biological system showed a stable and linear trend along the entire molecular dynamics run, suggesting good protein structural integrity (Figure 7D). The binding poses of biochanin A, sativanone, and pinocembrin exhibited RMSD values between 0.05 and 0.1 nm (Figure 7E), confirming the starting docking binding pose and stability. The nonbonded interaction energy of the target in complex with biochanin A, sativanone, and pinocembrin showed values of −140.6 ± 5.2 (−34.3 ± 1.2 kcal/mol), −137.3 ± 6.5 (−33.4 ± 1.5 kcal/mol), and −108.6 ± 4.9 kJ/mol (−26.4 ± 1.2 kcal/mol), respectively.

In silico results showed that biochanin A, sativanone, and pinocembrin were able to spontaneously bind to the K_Ca_1.1 channel showing Gibbs free energy values (ΔG) of −6.1, −5.8, and −5.4 kcal/mol, respectively. Despite the compounds sharing the same binding pocket (Figure 8), they formed a different interaction network, likely due to the differences in their structures and chemical–physical proprieties. Biochanin A formed hydrophobic interactions with Lys-397, Lys-458C, and Phe-461C; five hydrogen bonds with Lys-300C, Ser-383C, Glu-387C, Tyr-398C, and Lys-458; a π stacking with Tyr-398C (Figure 8A). Pinocembrin triggered four hydrophobic interactions with Lys-397C, Lys-458C, Glu-465C, and Tyr-467C; three hydrogen bonds with Tyr-398C, Glu-454C, and Lys-458C; a π stacking with Tyr-398C (Figure 8B). Sativanone was involved in four hydrophobic interactions with Lys-397C, Tyr-402C, Phe-461C, and Tyr-467C; a hydrogen bond with Gly-399C; a π stacking with Tyr-398C (Figure 8C).

The interaction network between K_Ca_1.1 channel binding pocket residues and the three flavonoids was further investigated by performing computational alanine-scanning mutagenesis, taking into account only the residues involved in hydrogen bonds. When considering the contribution of electrostatic energy, hydrogen bonds, Van der Waals force, and desolvation energy, a significant loss in the binding free energy of biochanin A was observed when Glu-387C (ΔΔG = −5.2 kcal/mol) and Tyr-398C (ΔΔG = −4.7 kcal/mol) were mutated to alanine. On the contrary, this contribution was negligible for pinocembrin and sativanone (Figure 9).

## 4. Discussion

The findings presented in this work demonstrated that *Dalbergia tonkinensis* represents a valuable source of vasoactive compounds that can be further developed to obtain drugs capable of targeting more than one pathway involved in the pathogenesis of hypertension. Biochanin A, for example, stimulated K_Ca_1.1 channels, blocked Ca_V_1.2 channels, showed a marked vasorelaxant activity, and bound to residues that can substantiate its effects on both channels.

With a few exceptions, most of the compounds assessed exhibited a more prominent vasorelaxant activity in aorta rings depolarised by moderate rather than high concentrations of K^+^. This pharmacological profile, characterizing orobol, pinocembrin, formononetin, biochanin A, sativanone, and partially 3′-*O*-methylviolanone, is distinctive of K^+^ channel openers. In fact, as previously reported by Gurney [47] and then confirmed in our laboratory [49], the vasodilating efficacy of K^+^ channel openers decreases when the extracellular concentration of K^+^ increases, due to the reduced chemical gradient that limits its efflux through the open channels. Consequently, membrane hyperpolarisation is weak, and the muscle relaxant effect of the compound is blunted. Accordingly, patch-clamp recordings demonstrated that biochanin A is a potent stimulator of I_KCa1.1_. To the best of our knowledge, this is the first study demonstrating the direct stimulatory activity of biochanin A on K_Ca_1.1 channels, previously hypothesised based on indirect evidence also in spontaneously hypertensive rats [50]. In fact, Au et al. [51] failed to detect any effect of the compound on these channels in porcine left anterior descending coronary myocytes, probably because they tested only the 1 µM concentration. Orobol and sativanone were ineffective, while formononetin, 3′-*O*-methylviolanone, and pinocembrin showed a weak to medium inhibitory activity. Pinocembrin and formononetin have previously been demonstrated to stimulate K_Ca_1.1 channels in rat aorta myocytes [52,53]. This apparent discrepancy can be explained either by the different vascular beds studied or by the more hyperpolarised V_h_ used, likely not adequate to avoid voltage-dependent K^+^ channel contamination of the aorta recordings. Tectorigenin-7-*O*-[β-D-apiofuranosyl-(1→6)-β-D-glucopyranoside], butin and, especially, ambocin showed a remarkable K_Ca_1.1 channel stimulatory activity that, however, did not translate to spasmolysis towards moderate but also high K^+^-induced contraction. Further experiments are necessary to clarify this apparent discrepancy.

Although the number of molecules analysed was limited, the structural similarity allowed the discovery of several structural requirements, fundamental or detrimental to the stimulatory activity. The isoflavanones structure gave rise to ineffective compounds. Among flavanones, the presence of a hydroxyl group at the C-5 position of ring A (eriodictyol) along with the absence of substituents on the B ring (pinocembrin) transformed the stimulator butin into an I_Ba1.2_ inhibitor. On the contrary, among isoflavones, the presence of a hydroxyl group at the C-5 position of ring A accompanied by a single substituent in the para position on the benzylic moiety of the B ring (i.e., biochanin A, ambocin, and tectorigenin-7-*O*-[β-D-apiofuranosyl-(1→6)-β-D-glucopyranoside]) favoured the stimulatory activity. The only exception was tectoridin, though its glycosylation pattern was different from that of the two effective and structurally similar glycosides. The 2,3-double bond was irrelevant to the stimulatory activity. Finally, the finding that glycosides were effective on K_Ca_1.1 channels confirms previous data obtained with (±)-naringenin and its glycoside naringin [54]. On the contrary, flavonoid modulation of Ca_V_1.2 and K_ir_6.1 channels was reduced or even vanished when one or more sugar moieties were present on the aglycone backbone/scaffold [55,56,57,58].

The S6/RCK linker in the K_Ca_1.1 channel is crucial both for channel activation and for modulator binding, as shown by Gessner et al. [59] through in vitro mutagenesis. Noticeably, in silico results showed that the three flavonoids bound with high affinity inside the K_Ca_1.1 channel binding pocket located close to this region. On the one hand, this observation supports the activity of biochanin A. On the other, however, it was at variance with the effects of pinocembrin and sativanone observed on I_KCa1.1_, the former being a weak inhibitor and the latter an ineffective agent. Unfortunately, the computational approach was characterised by a significant loss of pharmacodynamics information due to the low cover value of our target against the template in some protein regions and low confidence in their secondary structure prediction. This caused an alteration in the structural integrity of the channel during the cMD run, thus impeding a proper analysis of the complex formed with the three compounds. To explain this apparent discrepancy, however, the interaction network of the three flavonoids in complex with the target was analysed, followed by computational alanine-scanning mutagenesis to define the energy contribution of each K_Ca_1.1 channel-binding residue. According to the interaction network analysis, only biochanin A formed a large hydrogen bond network, consistent with its stimulation of I_KCa1.1_. Furthermore, the in silico alanine-scanning mutagenesis indicated that only biochanin A showed a significant decrease in the binding free energy when Glu-387C and Tyr-398C were mutated to alanine, abolishing their interactions. Thus, the different activity of pinocembrin and sativanone on I_KCa1.1_ was likely because they did not show any energy contribution against the Glu-387C and Tyr-398C residues. Taken together, these observations indicate that the Glu-387C and Tyr-398C residues play a key role in the protein–modulator interaction, thus representing novel *consensus* binding residues for K_Ca_1.1 channel modulators.

The two isoflavanones investigated were fairly active against high K^+^-induced contraction, which was mainly due to the extracellular Ca^2+^ influx through Ca_V_1.2 channels. These channels play a fundamental role in the regulation of vascular tone [60] and, therefore, represent crucial targets in antihypertensive therapy. The inhibition of these channels, in fact, is one of the standard therapeutic approaches to counteract the abnormal increase in blood pressure. The electrophysiology data pointed to sativanone, 3′-*O*-methylviolanone, pinocembrin, and biochanin A as effective Ca^2+^ antagonists, all characterised by a similar potency. Again, to the best of our knowledge, this is the first study demonstrating the direct inhibitory activity of pinocembrin and biochanin A on Ca_V_1.2 channels, previously hypothesised based on indirect evidence [50,52]. The structure–activity relationship analysis underlined the presence of an unsubstituted hydroxyl group in the C-7 position of ring A as essential for the Ca^2+^ antagonistic activity, all investigated 7-*O*-glycosides being weak inhibitors/stimulators or ineffective. Among the aglycones, I_Ba1.2_ inhibition was greater when one, two, or three methoxy groups were present on the B ring positioned in the C-3 position of the C ring (formononetin, biochanin A, sativanone, and 3′-*O*-methylviolanone) or when the B ring positioned in C-2 of the C ring was devoid of substituents (pinocembrin). The presence of the 2,3-double bond and the position of the B ring was irrelevant to the inhibitory activity (compare biochanin A with pinocembrin). Finally, the weak Ca^2+^ antagonism displayed by eriodictyol and formononetin was in line with previous observations obtained in aorta rings [50,61].

Interesting clues arise from the biophysical analysis of the effect of biochanin A on Ca_V_1.2 channels. The molecule shifted the steady-state inactivation curve to more negative potentials, indicating its capability to stabilize the channel in the inactivated state. Consistent with this hypothesis, biochanin A, like the two isoflavanones, sativanone and 3’-*O*-methylviolanone, sped up the inactivation kinetics of the I_Ba1.2_ (i.e., accelerated the transition from the open to the inactivated state). However, its efficacy was not reduced by the shift of the membrane potential to more hyperpolarised values, i.e., when many channels are in the resting, closed state, thus suggesting that the molecule can bind to channels both in the inactivated and resting state. On the contrary, neither the activation kinetics nor the steady-state activation curve was affected by biochanin A, indicating that the molecule did not modify the voltage sensitivity of the channel activation and the transition from the resting to the open state. Nevertheless, biochanin A caused a marked decrease in the window current, which is thought to be largely responsible for both the generation and regulation of vascular smooth muscle tone [62], which, in in vivo conditions, might be reduced by biochanin A. Finally, biochanin A-induced inhibition of I_Ba1.2_ was only partially reversible upon washout, thus limiting its direct use in clinics.

The patch-clamp findings provided a few insights into the mechanism(s) underpinning the flavonoid vasorelaxant activity. The spasmolytic effect of pinocembrin and the two isoflavanones sativanone and 3’-*O*-methylviolanone can be ascribed essentially to their Ca^2+^ antagonistic effect. The same did not apply to formononetin and orobol, as their Ca_V_1.2 channel inhibitory effect was weak. In biochanin A, the K_Ca_1.1 channel stimulation likely prevailed over the Ca_V_1.2 channel blockade, as the isoflavone was more effective on tissues depolarised with moderate concentrations of K^+^. Finally, butin, ambocin, and tectorigenin-7-*O*-[β-D-apiofuranosyl-(1→6)-β-D-glucopyranoside] did not show any vasorelaxant effect despite their marked K_Ca_1.1 channel stimulatory activity observed in single myocytes. While the access to the site of action of the latter two flavonoids might be prevented by the presence of the glycoside moiety, further experiments are needed to clarify the contradictory behaviour of butin. Tectoridin inactivity is complementary to the data obtained in the same preparation precontracted by phenylephrine [63].

The computational results showed that the three flavonoids bound spontaneously within a Ca_V_1.2 channel inhibitor binding pocket [27] with high stability and affinity, triggering a hydrophobic and polar interaction network. Remarkably, biochanin A was the only compound able to form a π-stacking interaction with Phe-1143 (pore forming), a key residue both for the binding of potent Ca_V_1.2 blockers and for channel inhibition [27]. The high stability of the binding poses of the three flavonoids as well as the high nonbonded interaction energy within the target binding pocket supports the strength of their interaction with the Ca_V_1.2 channel, likely responsible for a marked modification of the protein function. Altogether, structural and energy evidence provided by the cMD analysis were consistent with the in vitro results.

This work presents two limitations. First, disruption of the extracellular matrix to isolate single smooth muscle cells for patch-clamp recordings leads to the loss of focal contact tension that, in turn, considerably alters cell signalling systems [64]. Therefore, a phenomenon observed in isolated myocytes, though providing important insight into the drug’s mechanism of action, must be considered cautiously due to the absence of the physiological environment and always verified in intact tissues. The results here presented are consistent with a substantial contribution of both Ca_V_1.2 and K_Ca_1.1 channels, two of the main pathways controlling smooth muscle tone, analysed separately in intact vascular tissues simply varying the extracellular KCl concentration (see above) and quantified by electrophysiology experiments, to the vasorelaxant effect of the active compounds. However, muscle tone is an epiphenomenon controlled by several other pathways beyond the Ca_V_1.2 and K_Ca_1.1 channels, which may be affected by *Dalbergia* isolates. Second is the specificity of biochanin A. Smooth muscle and cardiac Ca_V_1.2 channels, in fact, exhibit only minor differences due to the alternative splicing loci [65]; thus, biochanin A might affect the electrical and mechanical activity of the heart. On the other hand, the possible activation of cardiac K_Ca_1.1 channels could have only beneficial effects, because their expression in the mitochondria of adult cardiomyocytes [66] plays a crucial role in cardioprotection [67]. Additional experiments are required to clarify these issues.

## 5. Conclusions

In conclusion, the present findings point to *Dalbergia tonkinensis* Prain as a valuable source of vasoactive compounds. Biochanin A, in particular, represents an interesting multitarget molecule, capable of simultaneously blocking Ca_V_1.2 channels and stimulating K_Ca_1.1 channels of vascular smooth muscle, thus leading to vasodilation. In this context, it is important to underline that cardamonin, a flavonoid precursor chalcone, is also endowed with the same bifunctional activity [68], suggesting that the “natural” flavonoid library could contain other molecules with similar multitarget activity. As these two targets are involved in the pathogenesis of hypertension [69,70], biochanin A might contribute to the development of novel antihypertensive drugs, nowadays necessary to improve both patient compliance and the successful therapy of pathology with a high socioeconomic impact. Recent findings [71] support this hypothesis.

## Figures and Tables

**Figure 1 molecules-27-04505-f001:**
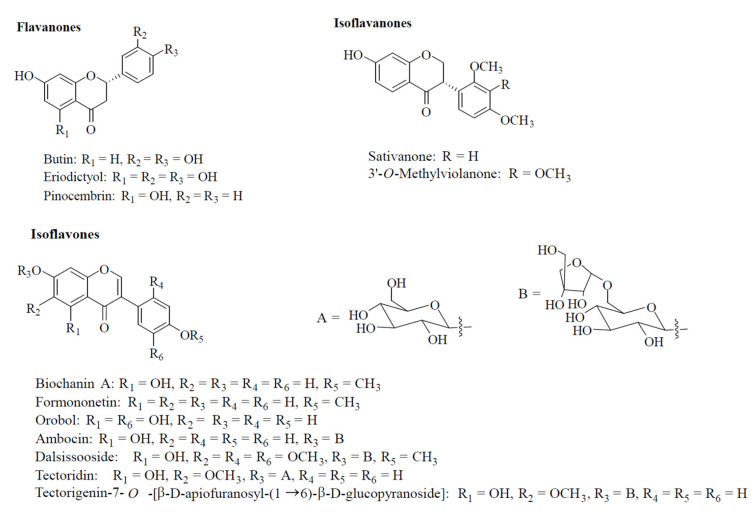
Structures of the molecules isolated from the heartwoods, roots, and leaves of *Dalbergia tonkinensis*. Compounds are grouped according to the subclass of flavonoids to which they belong.

**Figure 2 molecules-27-04505-f002:**
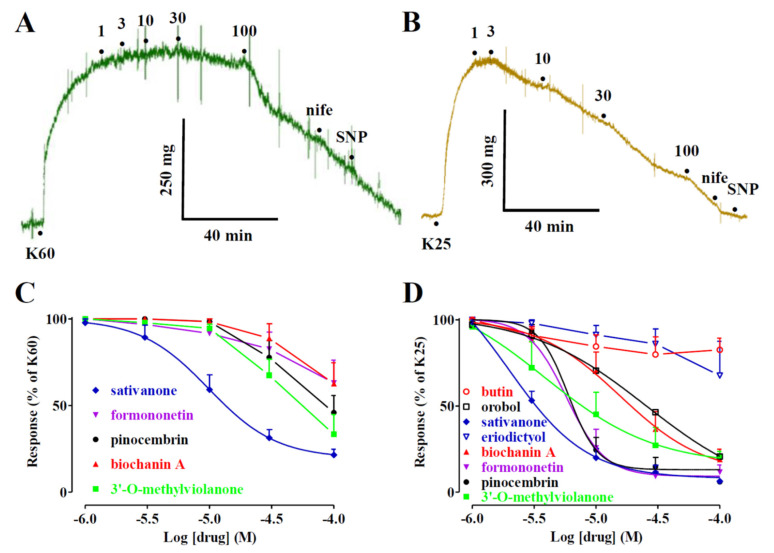
Effects of *Dalbergia* isolates on KCl-induced contraction of rat aorta rings. (**A**,**B**) Traces of vascular smooth muscle tension (representative of 4–6 similar experiments) showing the relaxation developed in response to cumulative concentrations of biochanin A (μM), added at the plateau of (**A**) 60 mM (K60) or (**B**) 25 mM KCl (K25) used to contract the preparations. The effect of 10 μM nifedipine (nife) and 100 μM sodium nitroprusside (SNP) is also shown. Horizontal bars indicate time (in min) and vertical bars muscle tension (expressed as mg). (**C**,**D**) Concentration–response curves for *Dalbergia* isolates in endothelium-denuded rings precontracted with either (**C**) 60 or (**D**) 25 mM KCl. In the ordinate scale, the response is reported as a percentage of the initial tension induced by KCl. Data points represent the mean ± SEM (*n* = 1–8).

**Figure 3 molecules-27-04505-f003:**
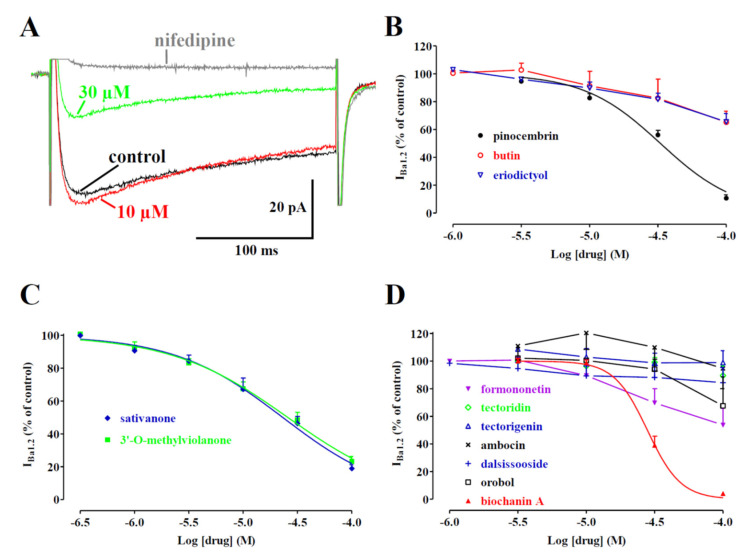
Effects of *Dalbergia* isolates on I_Ba1.2_ in single tail artery myocytes: (**A**) average traces (recorded from 5 cells) of I_Ba1.2_, elicited with 250 ms clamp pulses to 0 mV from a V_h_ of −50 mV, measured in the absence (control) or presence of cumulative concentrations of biochanin A; (**B**–**D**) concentration-dependent effects of (**B**) flavanones, (**C**) isoflavanones, and (**D**) isoflavones. On the ordinate scale, current amplitude is reported as a percentage of the value recorded just before the addition of the first concentration of *Dalbergia* isolate. The curves show the best fit of the points. Data points are the mean ± SEM (*n* = 1–6).

**Figure 4 molecules-27-04505-f004:**
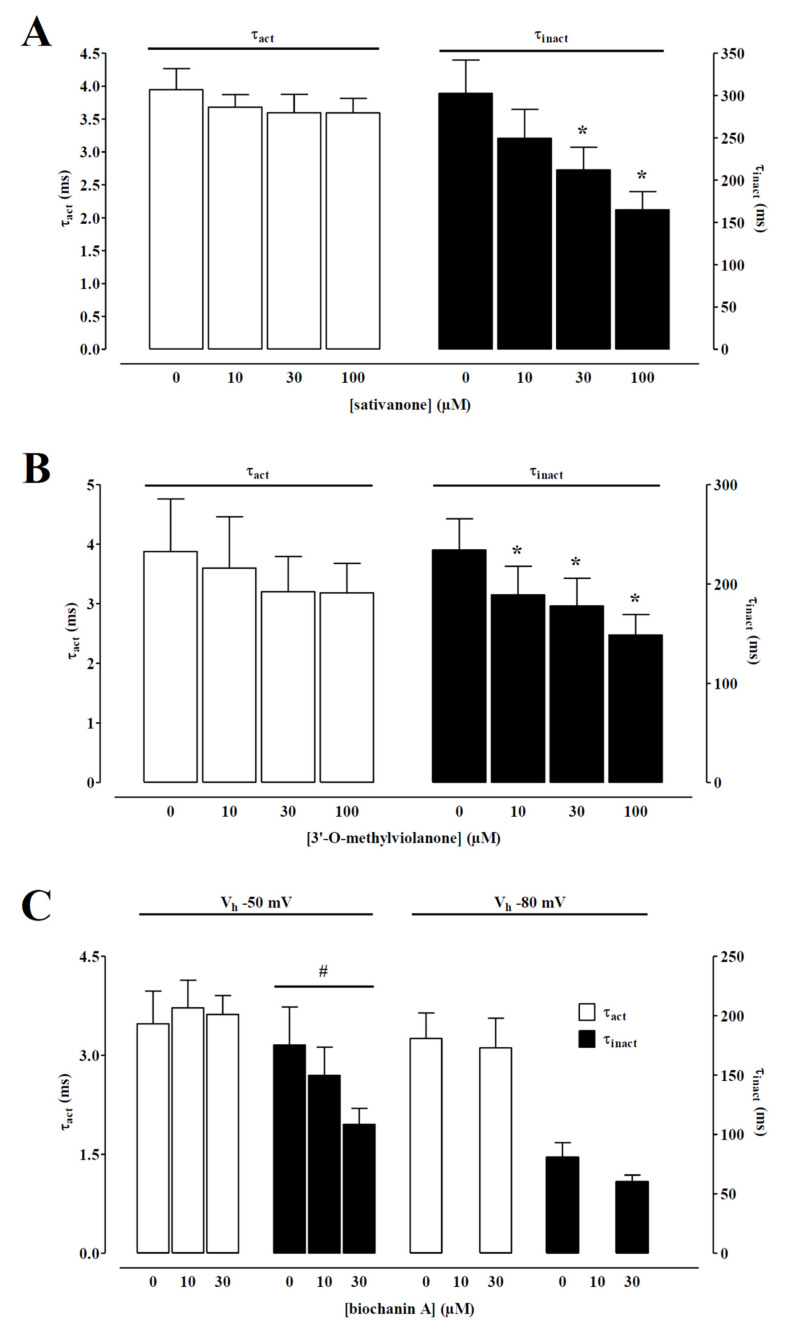
Effect of sativanone, 3′-*O*-methylviolanone, and biochanin A on I_Ba1.2_ kinetics of single tail artery myocytes. Time constant for activation (τ_act_) and inactivation (τ_inact_) measured in the absence or presence of various concentrations of (**A**) sativanone, (**B**) 3′-O-methylviolanone, and (**C**) biochanin A from a V_h_ of either −50 or −80 mV. Columns represent the mean ± SEM (*n* = 5–6). * *p* < 0.05 vs. control; one-way ANOVA and Dunnett post hoc test. # *p* < 0.05, one-way ANOVA.

**Figure 5 molecules-27-04505-f005:**
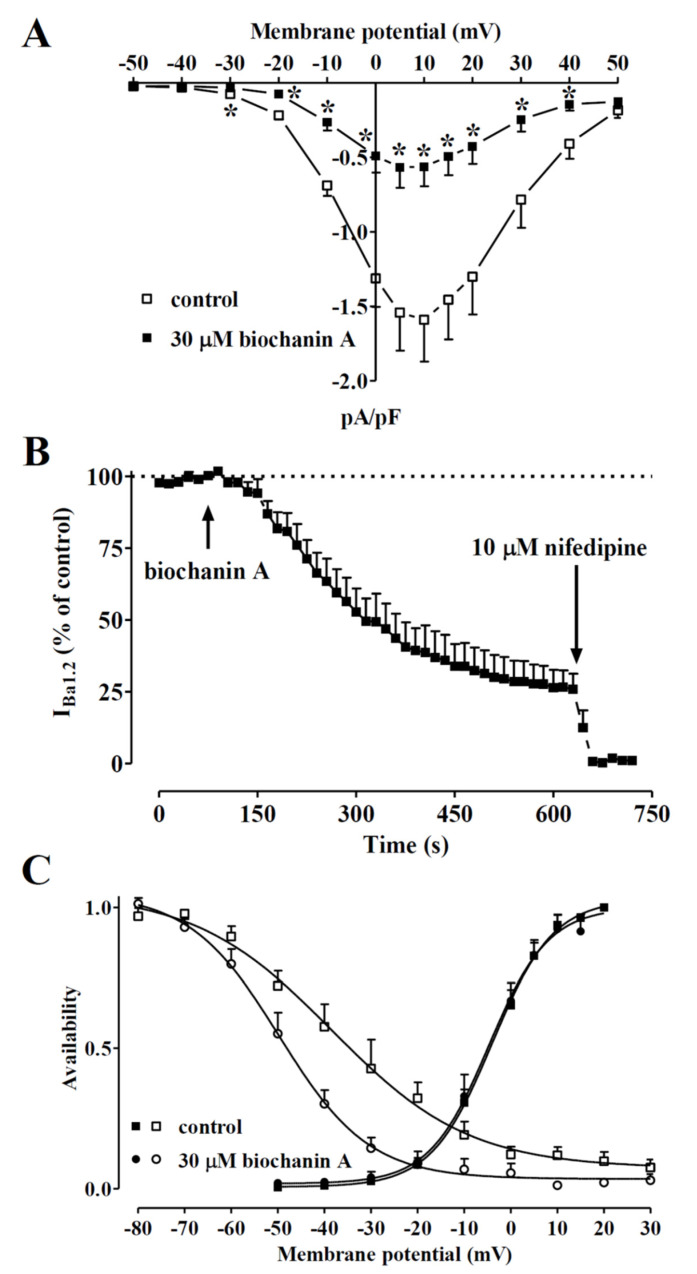
Voltage dependency of biochanin A-induced inhibition of I_Ba1.2_ in single tail artery myocytes. (**A**) Current–voltage relationships, recorded from a V_h_ of −50 mV, constructed before the addition (control) and in the presence of 30 µM biochanin A. Data points are the mean ± SEM (*n* = 5). * *p* < 0.05 vs. control, Student’s *t*-test for paired samples. (**B**) Time course of I_Ba1.2_ inhibition induced by 30 µM biochanin A. The drug was applied at the time indicated by the arrow, and peak currents were recorded during a typical depolarisation from −50 to 0 mV, applied every 15 s (0.067 Hz) and subsequently normalised according to the current recorded just before biochanin A addition. The effect of 10 µM nifedipine is also shown. Data points are the mean ± SEM (*n* = 5). (**C**) The effect of biochanin A on the voltage dependence of Ca_V_1.2 channel activation and inactivation. Steady-state inactivation curves were obtained using a double-pulse protocol. Once various levels of the conditioning potential had been applied for 5 s, followed by a short (5 ms) return to a V_h_ of −80 mV, a test pulse (250 ms) to 10 mV was delivered to evoke the current. The delay between the conditioning potential and the test pulse allowed for the full or near-complete deactivation of the channels, simultaneously avoiding partial recovery from inactivation. Steady-state inactivation curves in the absence (control) or presence of 30 µM biochanin A were fitted to the Boltzmann equation. Peak current values were used. The current measured during the test pulse was plotted against the membrane potential and is expressed as availability. Steady-state activation curves were obtained from the current–voltage relationships in panel A and fitted to the Boltzmann equation. Data points are the mean ± SEM (*n* = 5–6).

**Figure 6 molecules-27-04505-f006:**
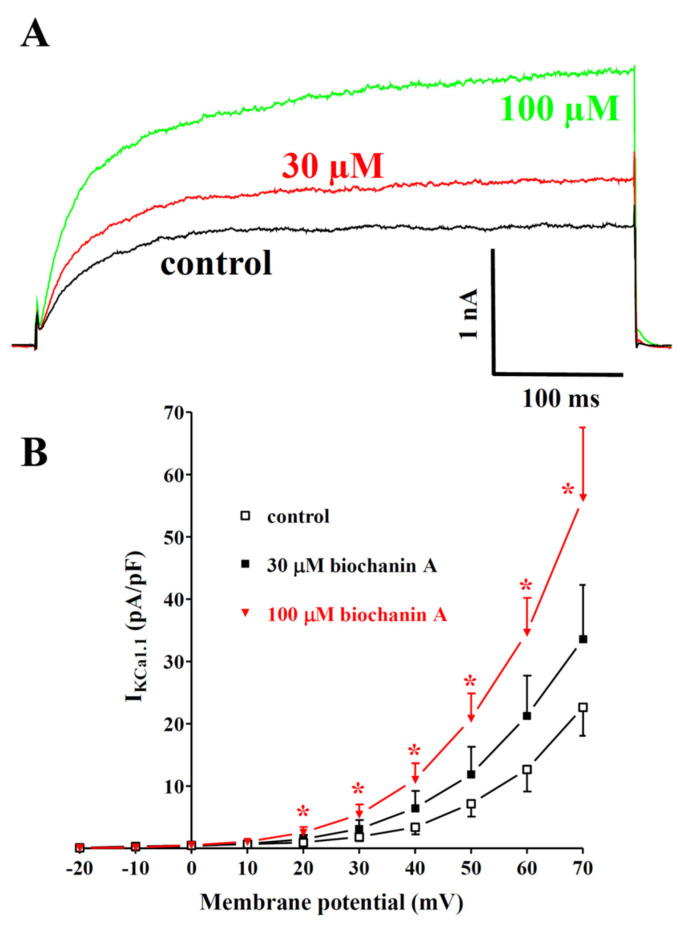
Effects of *Dalbergia* isolates on I_KCa1.1_ in single tail artery myocytes: (**A**) original recordings (average traces of 6 cells) of conventional whole-cell I_KCa1.1_ elicited with a 500 ms voltage step from V_h_ -40 to 70 mV, measured in the absence (control) and presence of 30 and 100 μM biochanin A; (**B**) current–voltage relationships obtained before the addition (control) and in the presence of various concentrations of biochanin A. On the ordinate scale, the response is reported as the current density in pA/pF. Data points are the mean ± SEM. * *p* < 0.05 vs. control; repeated measures ANOVA and Dunnett’s post hoc test.

**Figure 7 molecules-27-04505-f007:**
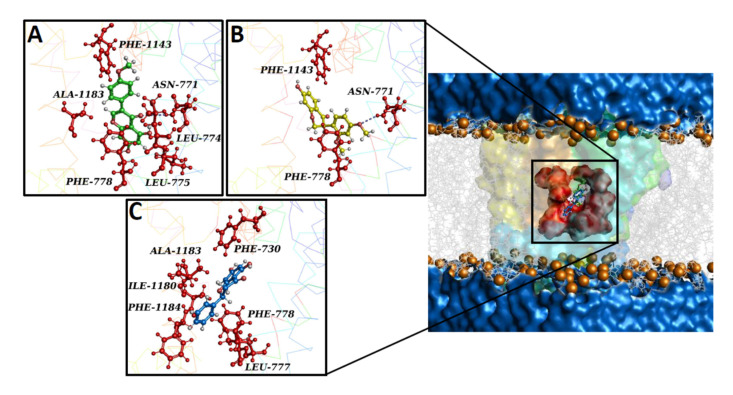
Overview of the Ca_V_1.2 channel docked with biochanin A, sativanone, and pinocembrin. The Ca_V_1.2 channel’s 3D structure is depicted with a multicolour transparent surface, while the binding pocket is reported in red. The bilayer is represented by grey lines, and some phospholipid heads are shown as orange spheres. The extracellular and cytoplasmatic side is shown as a blue surface. (**A**–**C**) Interaction network of (**A**) biochanin A, (**B**) sativanone, and (**C**) pinocembrin in complex with the Ca_V_1.2 binding residues (i.e., red balls and sticks) after the docking simulation. The hydrogen bond is represented as a purple dotted line. (**D**,**E**) Root mean square deviation (RMSD) profiles of biochanin A, pinocembrin, and sativanone. (**D**) RMSD profiles of the Ca_V_1.2 channel backbone in complex with biochanin A, pinocembrin, and sativanone. (**E**) RMSD profiles of the biochanin A, pinocembrin, and sativanone binding pose in complex with the Ca_V_1.2 channel. The RMSD trends are represented as coloured lines (see legend). RMSD (nm) and time (ns) values of the MD run are reported on the *y*- and *x*-axis, respectively.

**Figure 8 molecules-27-04505-f008:**
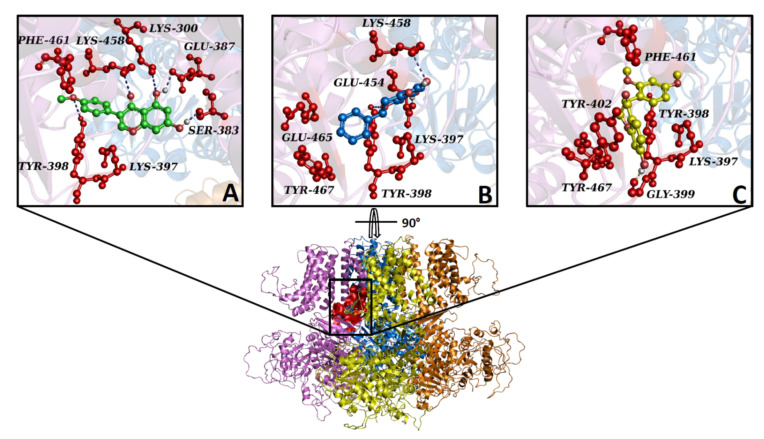
Overview of K_Ca_1.1 channels docked with biochanin A, pinocembrin, and sativanone. The K_Ca_1.1 channel’s 3D structure is depicted in a multicolour illustration, while the binding pocket is reported as a red surface. (**A**–**C**) Interaction network of (**A**) biochanin A, (**B**) pinocembrin, and (**C**) sativanone in complex with the K_Ca_1.1 binding residues (red balls and sticks) after the docking simulation. The hydrogen bond is represented as a purple dotted line. For clarity, only polar hydrogens are shown.

**Figure 9 molecules-27-04505-f009:**
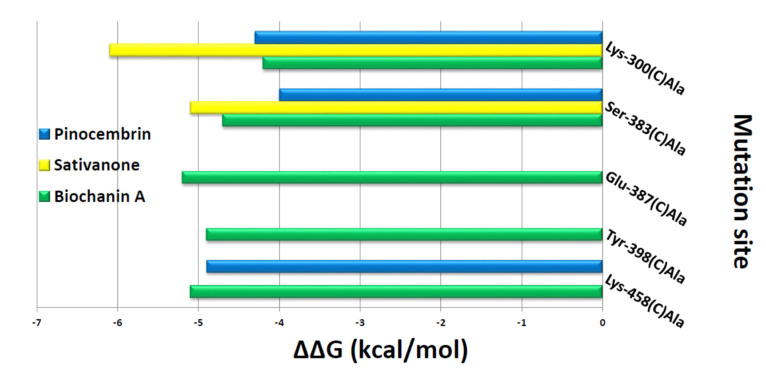
In silico alanine-scanning mutagenesis. Binding energy change values (ΔΔG = ΔG wild-type—ΔG Ala) obtained from the computational analysis of the K_Ca_1.1 channel in a complex with biochanin A, pinocembrin, and sativanone. ΔΔG values (kcal/mol) and the alanine-scanning mutagenesis of K_Ca_1.1 channel-binding residues (involved in hydrogen bonds) in complex with biochanin A, pinocembrin, and sativanone are reported on the *x*- and *y*-axis, respectively. Negative ΔΔG values indicate an unfavourable substitution for alanine in the relevant position.

**Table 1 molecules-27-04505-t001:** Effects of *Dalbergia* isolates on KCl-induced contraction in rat aorta rings and on rat tail artery myocyte K_Ca_1.1 and Ca_V_1.2 channel currents.

	Rat aorta Rings				Rat Tail Artery Myocytes		
Flavonoids					I_KCa1.1_		I_Ba1.2_
	25 mM KCl		60 mM KCl		30 µM	100 µM	IC_50_ (µM)
	IC_50_ (µM)	E_max_ (%)	IC_50_ (µM)	E_max_ (%)			
** *Flavanones* **							
Butin	N.D.	17.4 ± 6.8 (4)	N.D.	3.3 ± 2.2 * (6)	32.5 ± 33.7% (3)	73.5 ± 55.2% (3)	N.D.
Eriodictyol	N.D.	32.3 ± 19.9 (4)	N.D.	3.0 ± 3.0 (3)	-29.2% (1)	−21.1% (1)	N.D.
Pinocembrin	8.0 ± 1.1 (5)	94.0 (1)	N.D.	54.1 ± 9.9 (7)	-23.2 ± 15.4% (5)	−43.1 ± 18.2% (5)	32.7 ± 2.4 (5)
** *Isoflavanones* **							
Sativanone	4.2 ± 0.7 (5)	93.6 (1)	18.9 ± 4.7 * (8)	78.5 ± 3.2 (6)	14.5 ± 16.7% (6)	−5.2 ± 20.6% (6)	24.9 ± 5.5 (5)
3′-*O*-Methylviolanone	17.6 ± 8.3 (5)	79.4 (2)	N.D.	66.6 ± 11.3 (5)	−20.0 ± 13.1% (5)	−26.3 ± 11.1% (5)	27.9 ± 6.3 (5)
** *Isoflavones and isoflavone glycosides* **							
Biochanin A	23.4 ± 7.9 (6)	81.0 ± 6.0 (4)	N.D.	37.4 ± 12.1* (6)	51.4 ± 39.1% (6)	205.6 ± 111.6% (6)	28.2 ± 2.2 (5)
Formononetin	7.7 ± 1.6 (6)	90.7 ± 2.7 (6)	N.D.	36.9 ± 13.1* (6)	−3.6 ± 8.6% (5)	−12.7 ± 10.9% (5)	N.D.
Orobol	29.3 ± 11.6 (3)	79.3 ± 6.2 (3)	N.D.	3.0 (2)	0.8 ± 7.7% (4)	−5.1 ± 25.3% (4)	N.D.
Ambocin	N.D.	15.5 ± 11.5 (4)	N.D.	2.4 ± 2.1 (3)	74.0 ± 41.7% (4)	127.5 ± 82.3% (4)	N.D.
Dalsissooside	N.D.	6.4 ± 2.4 (3)	N.D.	2.1 ± 2.1 (3)	−19.3 ± 6.2% (5)	−32.5 ± 9.5% (5)	N.D.
Tectoridin	N.D.	18.4 ± 9.2 (3)	N.D.	2.0 ± 2.0 (3)	2.9 ± 6.2% (6)	2.9 ± 12.2% (6)	N.D.
Tectorigenin-7-*O*-[β-D-apiofuranosyl-(1→6)-β-D-glucopyranoside]	N.D.	3.5 ± 1.5 (3)	N.D.	0.6 ± 4.8 (3)	34.3 ± 11.3% (5)	58.8 ± 11.5% (5)	N.D.

Potency (expressed as IC_50_ value) and efficacy (E_max_, expressed as percent maximal relaxation) are mean ± S.E.M. (in parentheses the number of independent replicates). For I_KCa1.1_ data represent the percentage change of current amplitude caused by either 30 µM or 100 µM concentrations at 70 mV depolarization pulse. N.D.: not detectable. * *p* < 0.05 vs 25 mM KCl, Student’s *t* test for unpaired samples.

**Table 2 molecules-27-04505-t002:** Effects of *Dalbergia* isolates on the different settings analysed.

Flavonoids	Rat Aorta Rings		Rat Tail Artery Myocytes	
	25 mM KCl	60 mM KCl	I_KCa1.1_	I_Ba1.2_
** *Flavanones* **				
Butin	↔	↔	↑↑	↓
Eriodictyol	↓	↔	↔	↓
Pinocembrin	↓↓↓	↓↓	↓	↓↓↓
** *Isoflavanones* **				
Sativanone	↓↓↓	↓↓↓	↔	↓↓↓
3′-*O*-Methylviolanone	↓↓↓	↓↓	↓	↓↓↓
** *Isoflavones and isoflavone glycosides* **				
Biochanin A	↓↓↓	↓	↑↑↑↑↑↑↑	↓↓↓
Formononetin	↓↓↓	↓	↔	↓
Orobol	↓↓↓	↔	↔	↓
Ambocin	↔	↔	↑↑↑↑↑	↔
Dalsissooside	↔	↔	↓	↔
Tectoridin	↔	↔	↔	↔
Tectorigenin-7-*O*-[β-D-apiofuranosyl-(1→6)-β-D-glucopyranoside]	↔	↔	↑↑	↔

↓: inhibition; ↔: no effect; ↑: stimulation.

## Data Availability

All data are reported in the article.

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
