# Peer review of "Vietnamese Dalbergia tonkinensis: A Promising Source of Mono- and Bifunctional Vasodilators"

_molecules, 2022, doi:10.3390/molecules27144505_

Round 1
Reviewer 1 Report
The manuscript entitled "Vietnamese Dalbergia tonkinensis: a Promising Source of 2 Mono- and Bi-functional Vasodilators" is well-written. However, some clarifications are needed:
1. During the discussion, some aspects are missing:
-specificity of the ion channels effects, since it seems to be a non-specific regulation, how could this affect other ion channels, specifically cardiac ion channels, and thus undesired side effects.
-The effects were studied separated on both ion channels; however, both channels co-exist in the cells, would the effect of the compounds remain the same, opening K channels and the consequent hyperpolarization would anyway prevent Cav opening? how would the blocking effect on Cav play a role in the mechanism of action?
2. The authors indicate that in the phytochemical analysis, 35 flavonoids were isolated and identified by NMR analysis and literature comparison; however, no supplementary materials were provided, so I could not verify these claims.
Author Response
Manuscript ID molecules-1802012
Response to Reviewer #1
Comments and Suggestions for Authors
- During the discussion, some aspects are missing:
-specificity of the ion channels effects, since it seems to be a non-specific regulation, how could this affect other ion channels, specifically cardiac ion channels, and thus undesired side effects.
Response. This point has been considered and discussed in the revised manuscript (l. 733-739).
-The effects were studied separated on both ion channels; however, both channels co-exist in the cells, would the effect of the compounds remain the same, opening K channels and the consequent hyperpolarization would anyway prevent Cav opening? how would the blocking effect on Cav play a role in the mechanism of action?
Response. Disruption of the extracellular matrix to isolate single smooth muscle cells for patch-clamp recordings leads to the loss of focal contact tension that, in turn, considerably alters cell signalling systems (Ratz et al., Am J Physiol Cell Physiol, 2005;288:C769). Therefore, a phenomenon observed in isolated myocytes, though providing important insight into drug mechanism of action, has to be taken into account cautiously due to the absence of the physiological environment and always verified in intact tissues. CaV1.2 and KCa1.1 channels are two of the main pathways controlling vascular smooth muscle tone. As already stated in the previous version of the manuscript (now on l. 359-361), their contribution can be analysed separately in intact vascular tissues, simply varying the extracellular KCl concentration (Gurney, J Pharm Pharmacol, 1994;46:242). However, muscle tone is only an epiphenomenon controlled also by several other pathways beyond CaV1.2 and KCa1.1 channels. Based on the results obtained, these channels seem to contribute to the vasorelaxant effect of the active compounds. Only the electrophysiology experiments, however, can quantify their contribution to the final effect. This issue has been considered in the Discussion section (l. 721-732).
- The authors indicate that in the phytochemical analysis, 35 flavonoids were isolated and identified by NMR analysis and literature comparison; however, no supplementary materials were provided, so I could not verify these claims.
Response. The phytochemical analysis was performed in a former project. In the present article, only a selection of the compounds previously identified was assessed. The corresponding sentence has been modified (l. 350-352).
Reviewer 2 Report
The paper examines the effect of a collection of natural compounds directly on ion channels involved in hypertension and the contraction of aorta rings. The manuscript is challenging to follow due to the high number of compounds, systems studied and the variability in the observed effects. Although binding sites are suggested for the compounds, the study is rather descriptive, blocking/potentiation mechanisms or state-dependence are not investigated.
Major points:
1. Considering the high number of combinations of compounds and channels/effects tested, the authors should include a summary table showing what effect, if any, each compound had on each of the tested systems.
2. Nothing is shown or stated about the reversibility of the effects, like aorta ring relaxation or channel block. It is an important pharmacological property, which must be addressed. For example, in an experiment like Fig 5B, washout should be started at the point where nifedipine was added to show how the amplitude changes upon removal of the compound.
3. Which gating state of the channels was used for modeling? The authors should include thoughts about the sidedness and state-dependence of the block / potentiation, preferably supported by experimental results. The mechanisms are not addressed at all. For example, the left-shift of the SSI curve of Cav1.2 by biochanin-A is not discussed, neither is the effect of compounds on the rate of inactivation.
4. What was the point of using another holding potential (-80 mV)? Why was it only used for one compound?
5. It would be important to show with paxilline or IbTx that the current shown in Fig.6 is indeed BK current.
Minor:
1. It is not clear what the curves in Fig 2 represent. What is plotted as a function of time? The scale bar shows “mg”, which a unit of mass. It is explained neither in the Methods nor in the text or the legend. Please clarify.
2. What is the advantage of using barium as the charge carrier for CaV1.2 channels? The authors should include an explanatory sentence to rationalize this choice.
3. Fig 4A Round values on the axes would be more pleasing to the eye (e.g. 1.0, 2.0 … instead of 0.9, 1.8 …, likewise going by 50s or 100s rather than 70s on the right).
4. The design of Fig 4 is chaotic. Why use different arrangement and colors for panel C vs. panels A and B? Why use asterisks for panel B and indicate the p value on panel C? This should be made more consistent.
5. “Steady-state inactivation curves, recorded from a Vh of -80 mV… “ SSI curves are generally obtained by stepping to the same depolarizing pulse from different holding potentials, so how should this sentence be interpreted?
6. Resolution of Fig 7 is very poor, lower panels are not legible. In the upper panel on the right, the location of the binding pockets within the channel structure is not very well visible. The zoom level should be set between the panels on the left and the one on the right to clearly demonstrate where within the channel protein the molecules bind.
Author Response
Manuscript ID molecules-1802012
Response to Reviewer #2
Comments and Suggestions for Authors
Major points:
- Considering the high number of combinations of compounds and channels/effects tested, the authors should include a summary table showing what effect, if any, each compound had on each of the tested systems.
Response. A new Table 2, summarizing the effects of the compounds on rat aorta ring and tail artery myocyte settings, has been included in the revised manuscript (p. 14).
- Nothing is shown or stated about the reversibility of the effects, like aorta ring relaxation or channel block. It is an important pharmacological property, which must be addressed. For example, in an experiment like Fig 5B, washout should be started at the point where nifedipine was added to show how the amplitude changes upon removal of the compound.
Response. The effect of washout on biochanin A-induced inhibition of IBa1.2 is now stated in the Result section (l. 434-435) and discussed (l. 696-697).
- Which gating state of the channels was used for modeling? The authors should include thoughts about the sidedness and state-dependence of the block / potentiation, preferably supported by experimental results. The mechanisms are not addressed at all. For example, the left-shift of the SSI curve of Cav1.2 by biochanin-A is not discussed, neither is the effect of compounds on the rate of inactivation.
Response. The gating state of the channels used in the in silico study is now indicated in the revised manuscript (l. 300-304, and 304-311). We apologize to the Reviewer: the discussion of the biophysical analysis, previously missing, is now included in the Discussion section (l. 682-696).
- What was the point of using another holding potential (-80 mV)? Why was it only used for one compound?
Response. The use of a more hyperpolarised potential (i.e., Vh=-80 mV) was associated with the steady-state inactivation protocol, where all the channels have to be in the resting state before the various conditioning pulse are applied. This allowed the comparison of the effects of biochanin A at two Vhs, the more depolarised one (i.e., -50 mV) being characterised by a percentage of channels in the inactivated state (see Figure 5C). The comparison provided evidence of the channel state ideal for the binding of biochanin A. This is now clearly stated in the revised manuscript (l. 687-690). The reason it was assessed only for biochanin A is that it is the most interesting compound that emerged from this screening.
- It would be important to show with paxilline or IbTx that the current shown in Fig.6 is indeed BK current.
Response. These data have been already published in Saponara et al. (2006; Figure 3a) and Iozzi et al. (2013; Figure 2a-c), two references already cited in the previous version of the manuscript.
Minor:
- It is not clear what the curves in Fig 2 represent. What is plotted as a function of time? The scale bar shows “mg”, which a unit of mass. It is explained neither in the Methods nor in the text or the legend. Please clarify.
Response. The traces shown in Figure 2A,B were already described in the previous version of the manuscript and in the revised one (l. 359-361 and l. 377-382). To address this point, however, the corresponding legend has been modified.
- What is the advantage of using barium as the charge carrier for CaV1.2 channels? The authors should include an explanatory sentence to rationalize this choice.
Response. A new sentence is now included in the revised manuscript to rationalize the choice of Ba2+ as the charge carrier (l. 238-241).
- Fig 4A Round values on the axes would be more pleasing to the eye (e.g. 1.0, 2.0 … instead of 0.9, 1.8 …, likewise going by 50s or 100s rather than 70s on the right).
Response. Done.
- The design of Fig 4 is chaotic. Why use different arrangement and colors for panel C vs. panels A and B? Why use asterisks for panel B and indicate the p value on panel C? This should be made more consistent.
Response. The colours in Figure 4C have been changed. The asterisks show a statistically significant difference vs control according to the Dunnett post-hoc test. In panel C, one-way ANOVA indicated a statistically significant difference between the groups that, however, was not followed by a statistically significant difference of any group vs control. To clarify this point panel C has been modified and a new sentence has been included in the corresponding legend (l. 424).
- “Steady-state inactivation curves, recorded from a Vh of -80 mV… “ SSI curves are generally obtained by stepping to the same depolarizing pulse from different holding potentials, so how should this sentence be interpreted?
Response. The legend to Figure 5C has been modified and now describes the method followed to construct the steady-state inactivation curve (l. 465-471).
- Resolution of Fig 7 is very poor, lower panels are not legible. In the upper panel on the right, the location of the binding pockets within the channel structure is not very well visible. The zoom level should be set between the panels on the left and the one on the right to clearly demonstrate where within the channel protein the molecules bind.
Response. Figure 7 has been revised according to the Reviewer’s suggestions.
This manuscript is a resubmission of an earlier submission. The following is a list of the peer review reports and author responses from that submission.